nanotechnology

immobilization, bioconjugate, gold nanostars, glucose oxidase, nanobiosensor

**Authors for correspondence:**
Masauso Moses Phiri
e-mail: missiphiri@gmail.com
Barend Christiaan Vorster
e-mail: chris.vorster@nwu.ac.za

This article has been edited by the Royal Society of Chemistry, including the commissioning, peer review process and editorial aspects up to the point of acceptance.

# Facile immobilization of glucose oxidase onto gold nanostars with enhanced binding affinity and optimal function

Masauso Moses Phiri, Danielle Wingrove Mulder, Shayne Mason and Barend Christiaan Vorster

Centre for Human Metabolomics, North-West University, Potchefstroom Campus, Private Bag X6001, Potchefstroom, South Africa

MMP, 0000-0001-7653-7988; DWM, 0000-0002-6970-7392; SM, 0000-0002-2945-5768; BCV, 0000-0003-2371-288X

Gold nanoparticles provide a user-friendly and efficient surface for immobilization of enzymes and proteins. In this paper, we present a novel approach for enzyme bioconjugation to gold nanostars (AuNSs). AuNSs were modified with L-cysteine (Cys) and covalently bound to *N*-hydroxysulfosuccinimide (sulfo-NHS) activated intermediate glucose oxidase (GOx) to fabricate a stable and sensitive AuNSs–Cys–GOx bioconjugate complex. Such a strategy has the potential for increased attachment affinity without protein adsorption onto the AuNSs surface. Good dispersity in buffer suspension was observed, as well as stability in high ionic environments. Using the AuNSs–Cys–GOx bioconjugates showed greater sensitivity in the measuring of low concentrations of glucose based on plasmonic and colorimetric detection. Such a novel approach for enzyme immobilization can lead to AuNSs–Cys–GOx bioconjugate complexes that can be used as catalytic nanodevices in nanobiosensors based on oxidases in biomedical applications.

## 1. Introduction

One of the many advantages that gold nanoparticles offer to biosensors is a user-friendly and efficient surface for immobilizing enzymes [1]. The use of gold nanoparticles in biosensors—nanobiosensors—as signal transducers is a promising alternative to traditional detection techniques used in clinical diagnosis. Simplicity and cost-effectiveness in fabrication, sensitivity in signal transduction as well as easy readout platforms are some of the excellent properties of nanobiosensors [2]. Colloidal gold has high

biocompatibility and surface energy that enables immobilized enzymes to retain their bioactivity. They also increase enzyme loading as opposed to bulk materials where enzyme adsorption usually leads to protein denaturation and decreased performance [3,4].

An important prerequisite to clinical use of nanoparticles is their surface functionalization with biorecognition molecules. This functionalization aids in maintaining the stability of the biorecognition molecules and nanoparticle bioconjugates in biological matrices so as to preserve their functionality [4–6]. Enzyme immobilization onto colloidal gold nanoparticles enables more freedom of orientation for the attached enzyme with less probability of covering the active site [7]. Furthermore, immobilized enzymes increase in stability to heat and fluctuations in the chemical environment [8,9], although some of their properties such as the Michaelis constant ($K_m$) or optimum pH value may be changed in the process [4,10]. Thus, there is need for optimal strategies for enzyme immobilization in order to maximize assay sensitivity, selectivity, reproducibility and stability in biological samples [4–6].

The two main methods for enzyme immobilization on gold nanoparticles are physical and chemical adsorption. Physical methods employ weak interactions between the particle surface and enzyme, while in chemical methods, covalent bonds are formed between the enzyme and particle surface [3]. Physical adsorption is a quick and simple technique for attaching enzymes in biosensors. Despite the benefits of speed and simplicity that this method offers, there are disadvantages such as undesirable enzyme orientations, covering of active site and decreased enzyme functionality [7]. Interestingly, it is reported that direct adsorption is still being routinely used for conjugation of proteins to nanoparticles with reports of better binding activities for antibodies [11–13].

Chemical adsorption method of enzyme immobilization, on the other hand, involves direct covalent binding between the enzyme and the colloidal gold surface. These coupling chemistries aid in controlling the orientation of the immobilized protein onto the gold nanoparticles surfaces [7,13,14]. Enzyme immobilization to colloidal gold nanostructures is achieved by the use of cross-linker molecules of different lengths. The linker molecule gives the enzyme greater mobility, thereby enhancing its bioactivity, compared to that of a directly coupled immobilization [3,15]. Enzymes may be modified to have reactive groups that are useful for conjugation with appropriately functionalized gold nanoparticles [1,3]. To accomplish this, homobifunctional or heterobifunctional linker molecules are used to covalently couple some chemical target group on the enzyme and a resultant terminal reactive group that can cross-link with the supporting surface [15].

Glutaraldehyde is one of the most commonly used homobifunctional cross-linkers that contain an aldehyde group at both ends of a 5-carbon chain. It primarily reacts with amine groups with more than one mechanism of reaction. It is able to cross-link two molecules with amine groups and form stable bonds [15–17]. Another most popular cross-linker is (1-ethyl-3-(3-dimethylaminopropyl)carbodiimide hydrochloride (EDC) [18]. It is probably the most frequently used cross-linking agent of all. It is used for conjugating biological substances containing carboxylates and amines. It is mostly used along with *N*-hydroxysuccinimide (NHS) or the water-soluble *N*-hydroxysulfosuccinimide (sulfo-NHS) in particle and surface conjugation procedures [15,18]. These two cross-linkers account for most of the covalent enzyme–nanogold conjugation procedures encountered in the literature [3,4].

Six ideal conditions for immobilized enzymes onto nanoparticles are well delineated by Sapsford *et al.* [1]: (i) a high ratio of enzymes per nanoparticle to increase binding and interaction with target analyte; (ii) control over orientation of the enzyme attached to the nanoparticle, so that the active site is uncovered; (iii) control over relative separation distance between the enzyme and the nanoparticle; (iv) control over attachment affinity of bioconjugates; (v) maintenance of optimal function and activity of both the enzyme and nanoparticles; and, lastly, (vi) ability to be reproduced in a facile manner with other biomolecules to be immobilized.

Cysteine (Cys) is known to strongly bind to gold surfaces via the thiol group and form self-assembled monolayers [19,20]. In a study of the structural and bonding evolution in Cys–gold cluster complexes, the thiol moiety is reported to be a very effective site for interaction with gold nanoparticles in aqueous medium as observed from a number of techniques, such as UV–vis, Fourier transform infrared (FTIR), Raman and $^1$H-NMR spectroscopy [19,20]. Gold nanostructures thus functionalized with thiol ligands tend to drastically reduce non-specific protein adsorption on their surfaces [18,19,21–23]. Cys has previously been used together with glutaraldehyde as a support for enzyme immobilization without adsorption onto the surface with thermal and assay stability [24]. The disadvantages of using glutaraldehyde for enzyme conjugation reactions are: it is a very hazardous chemical, has a complex reaction mechanism, requires other harmful chemical reagents in the reaction process such as sodium cyanoborohydride, requires high pH of greater than 9 which may cause nanoparticle aggregation, and the cross-linking is difficult to reproduce and scale up [15–17]. Hence, there is need for optimal enzyme immobilization methods onto gold nanoparticles modified with Cys

that can prevent adsorption onto the surface, has thermal and assay stability, are simple to accomplish, use environmentally friendly chemicals and are easily reproducible.

In this study, a bioconjugation approach for attachment of enzymes to gold nanostars (AuNSs) was devised. AuNSs were the nanosensors of choice for signal transduction based on the localized surface plasmon resonance (LSPR). The LSPRs are determined by the shape of the nanoparticles' width, position and number [25]. A common feature of LSPRs for nanostars is their location at lower energy compared to nanospheres [26]. For example, gold nanospheres with the size of 2–50 nm show only one plasmon band centred at about 520 nm, while for nanostars, the plasmon band is red-shifted and more intense, and typically centred around 650–900 nm [26–29]. The trending way of late has been to use different morphologies and compositions of nanostructures, such as AuNSs, as a way to tune the LSPR properties of the nanosensors [30] for greater sensitivity [31]. Lastly, AuNSs also provide a larger surface area for enzyme immobilization with potential for higher load of enzymes per nanoparticle compared to smaller nanospheres [1].

The approach described here for enzyme immobilization to AuNSs brings together the use of Cys and EDC/sulfo-NHS to create an optimal conjugation that would prevent protein adsorption onto the surface of the particles and offer some relative separation distance between enzyme and nanoparticle. The bioconjugation approach is facile, easily reproducible, used simple chemistries with non-hazardous chemicals and generated stable and sensitive bioconjugates with increased attachment affinity without protein adsorption onto the AuNSs surface. This was accomplished by first functionalizing AuNSs with Cys. Secondly, glucose oxidase (GOx) was modified with EDC/sulfo-NHS to increase the stability and solubility of active esters intermediate, and to increase the conjugation yield. Thirdly and lastly, the cysteine-modified AuNSs (AuNSs–Cys) and NHS-terminated GOx were covalently coupled together to form AuNSs–Cys–GOx bioconjugates. Consequently, stable AuNSs–Cys–GOx bioconjugates were generated as proposed in scheme 1. The AuNSs–Cys–GOx bioconjugates synthesized by this approach were assessed for stability and exploited for glucose detection sensitivity in a nanobiosensor via an enzymatic assay comparison.

# 2. Material and methods

## 2.1. Materials and instrumentations

Hydrochloroauric acid (HAuCl$_4$), glucose oxidase (GOx), trisodium citrate, silver nitrate (AgNO$_3$), ascorbic acid, sodium chloride (NaCl), polyvinylpyrrolidone (PVP) (molecular weight 10 000), hydrochloric acid (HCl), glucose, 2-(N-morpholino)ethanesulfonic acid (MES) at pH 6, N-(3-Dimethylaminopropyl)-N′-ethylcarbodiimide hydrochloride (EDC), N-Hydroxysuccinimide (NHS), Cys and 1× phosphate-buffered saline (PBS) at pH 7.4, Pur-A-Lyzer Midi 3500 Dialysis Kit were all purchased from Sigma-Aldrich, South Africa. All glassware was stripped with Aqua Regia prior to use for synthesis. Ultra-pure water (ddH$_2$O) was pre-prepared with a Milli-Q ultra-pure system (18.2 MΩ cm$^{-1}$).

## 2.2. Preparation of AuNSs–Cys–GOx bioconjugates

A recently published method for the synthesis of seedless silver nitrate and ascorbic acid-assisted AuNSs by Phiri *et al.* [32] was followed. Briefly, 10 ml of ddH$_2$O was acidified with 10 µl of 1 M HCl. Thereafter, 50 µl of 100 mM ascorbic acid was added under mild stirring. To the mixture, 50 µl of 50 mM HAuCl$_4$ was added. Shortly and rapidly (within 30 s), 50 µl of 10 mM AgNO$_3$ was added to the solution which resulted in a deep blue colour change within a few seconds. Finally, 500 µl of 2.5 mM PVP was added to the mixture. The prepared AuNSs were cleaned up by centrifugation for 90 min at 3000$g$. The pellet was then recovered and resuspended in 2 ml of ddH$_2$O. The subsequent AuNSs–Cys were prepared by adding 100 µl of 0.02 mM Cys to 2 ml of PVP-stabilized AuNSs at pH 7 and left to incubate on a rotator for 3 h. The final mixture of Cys-modified AuNSs was dialysed in 0.8 ml Pur-A-Lyzer tubes using an in-house non-equilibrium dialysis system to remove excess unbound Cys, and re-dispensed in 1 ml of PBS. The chemical modification of enzyme was prepared by adding 250 mM of freshly prepared EDC/sulfo-NHS to 1 ml of GOx (5 mg ml$^{-1}$) in MES buffer (10 mM, pH 6) and allowed to react for 2 h. The excess EDC/sulfo-NHS molecules were removed by dialysis, as described above, from the modified enzymes. Finally, the conjugation of the AuNSs–Cys–GOx bioconjugates was accomplished by pipetting 500 µl of EDC/sulfo-NHS-modified enzymes and was added to 2 ml of AuNSs–Cys and incubated overnight in the fridge. Thereafter, the mixture was centrifuged at 3000$g$ for 30 min to remove any unbound enzymes. The AuNSs–Cys–GOx bioconjugates were then resuspended in MES buffer and stored at 4°C until usage.

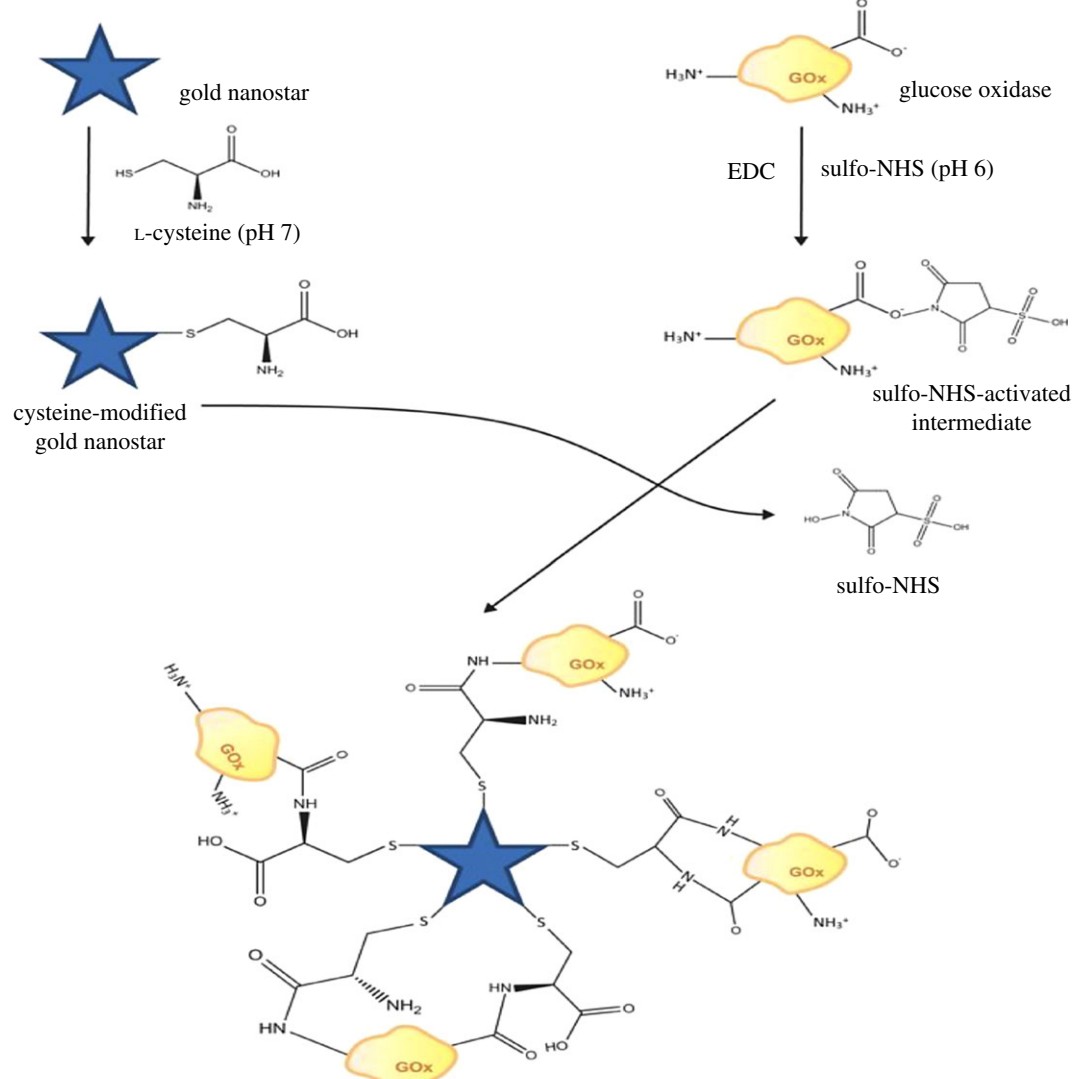

**Scheme 1.** Procedure and plausible covalent attachments of GOx to Cys-modified AuNSs.

## 2.3. Characterizations and instrumentations

[1]H-NMR analyses of the samples in fabrication stages were done according to the method by Venter *et al.* [33]. Six hundred microlitres of samples were measured at 500 MHz on a Bruker Avance III HD NMR spectrometer equipped with a triple-resonance inverse [1]H[[15]N,[13]C] probe head and *x*, *y* and *z* gradient coils. [1]H spectra were acquired as 128 transients in 32 K data points with a spectral width of 12 002 Hz. Fourier transformation and phase and base line correction were done automatically. Software used for NMR processing was Bruker Topspin (v. 3.5). Bruker AMIX (v. 3.9.14) was used for metabolite identification [34]. UV–vis spectroscopy analyses were carried out by spectral scanning (400–990 nm) on an HT Synergy (BioTEK) microplate reader. The transmission electron microscopy (TEM) analyses were performed on a Tecnai F20 high-resolution transmission electron microscope (HR-TEM) at an accelerating voltage of 200 kV. Samples for TEM were prepared by applying 20 µl of nanoparticle suspension onto carbon 200 mesh copper grids (Agar Scientific), followed by drying overnight prior to imaging.

## 2.4. Stability of AuNSs – Cys – GOx bioconjugates

Using a method applied by Rodríguez-Lorenzo *et al.* [31] to prove the stability of protein-modified AuNSs, the stability of the AuNSs–Cys–GOx bioconjugates was investigated in solutions containing high ionic strength. The AuNSs–Cys–GOx bioconjugates were centrifuged and resuspended in 300 mM NaCl.

UV–vis spectroscopy was used to observe if there was any aggregation that would be evident by shifts in the LSPRs of the AuNSs to longer wavelengths and flattening of the absorption spectra.

## 2.5. Glucose sensing using AuNSs–Cys–GOx bioconjugates

Glucose determination was carried out to test the feasibility of the AuNSs–Cys–GOx bioconjugates' application as catalytic nanodevices in a nanobiosensor. A previously optimized method for glucose sensing using differently functionalized AuNSs–GOx [32] was followed for the newly prepared AuNSs–Cys–GOx bioconjugates. Briefly, a range of glucose concentration standards (0.2–2 mM) were added to different reaction wells in a 96-well plate. Each 200 µl reaction solution contained 30 µl AuNSs–Cys–GOx bioconjugates, 1 mM MES buffer and a specified concentration of glucose added to it. The mixture was incubated for 1 h at 37°C after which 12 µl detection solution was added. The detection solution was a combination of 0.1 mM $AgNO_3$ and $NH_3$ (10 mM)/NaOH (25 mM) equi-volume mixture. Three comparison assays were done to assess if the AuNSs–Cys–GOx bioconjugates offered any advantage in biosensing: (i) AuNSs only without the addition of any GOx to it, as assay controls; (ii) AuNSs with 5 µl GOx added to the reaction solutions; and, lastly, (iii) AuNSs–Cys–GOx bioconjugates in solution as catalytic nanodevices and signal transducers for glucose determination. The detection of glucose was assessed based on shifts in the LSPR peaks on the spectrophotometer and by colour change of the solutions. The mechanism of detection was via biocatalytic enlargement of AuNSs through surface coating of $Ag^0$ after its reduction by hydrogen peroxide. The hydrogen peroxide itself is a product of the oxidation of glucose in the presence of glucose oxidase and molecular oxygen. These chemical equations depict the reaction:

$$\beta\text{-}D\text{-glucose} + O_2 + H_2O \xrightarrow{\text{GOx}} D\text{-gluconic  acid} + H_2O_2$$
$$\text{and}\quad 3H_2O_2 + Ag^+ \xrightarrow{\text{AuNSs}} Ag^0 + 3H_2O + \frac{3}{2}O_2.$$

# 3. Results and discussion

Scheme 1 illustrates and summarizes the proposed procedure for the AuNSs–Cys–GOx bioconjugation approach. After the removal of excess PVP from the AuNSs, AuNSs–Cys was obtained via ligand exchange reaction at pH 7. At this neutral pH, the AuNSs–Cys system is known to form stable Au–S structures [19]. The thiol bond between Cys and AuNSs leaves a secondary amine and a carboxylate group at the terminal end of the molecules. The enzyme was chemically modified by incubation in EDC/sulfo-NHS in MES buffer at pH 6 to form an active ester before conjugation with an amide- or carboxylate-containing group [35]. EDC was used in order to react with a carboxylate group on the enzyme to form an active ester leaving group. Sulfo-NHS was added to the EDC reaction to increase the solubility and stability of the active intermediate, which ultimately reacts with the attacking amine/carboxylate groups from the Cys. The advantage of EDC/sulfo-NHS coupled reactions is that they are highly efficient and tend to increase the yield of conjugation significantly over that obtainable solely with EDC [35]. The Cys-modified AuNSs and NHS-terminated GOx could couple in a number of plausible conjugation reactions (scheme 1) to form amide bond linkages. Thus, a GOx monolayer was covalently immobilized on the surface of AuNSs in such a way as to avoid non-specific binding of the protein, and to potentially increase both affinity and stability of GOx attachment of the AuNSs [15,18,24,36].

[1]H-NMR spectra of the AuNSs–GOx (figure 1) show discernible shifts and splitting on the modified molecules compared to pure standards. Spectrum (figure 1a) shows free Cys compared to AuNSs–Cys (figure 1b). In the region between 3.0 and 4.5 ppm, where representative peaks for Cys exist at 3.25 ppm ($CH_2$) and 4.10 ppm (CH), the AuNSs–Cys shows that these peaks are drawn together at 3.75 ppm, indicating a shift most likely due to the specific interaction of gold with the sulfur. There is probably an increase in electron density and plausible formation of hydrogen bond between the gold surface-bound Cys molecules and the neighbouring bound Cys of the next gold particles, also observed in other studies [37–39]. Spectrum (figure 1c) shows the peaks for the NHS-terminated GOx. The NHS-terminated GOx's spectrum shows the shifts and different peaks which are due to the esterification of the GOx with EDC/sulfo-NHS—indicating a successful chemical modification of the enzyme. Spectrum (figure 1d) shows the bioconjugation of AuNSs–Cys with NHS-terminated GOx with slightly shifted peaks for the ester-activated enzyme and the AuNSs–Cys, all the three essential molecules in the bioconjugation approach indicating a successful AuNSs–Cys–GOx conjugation. Other recommended spectroscopic techniques

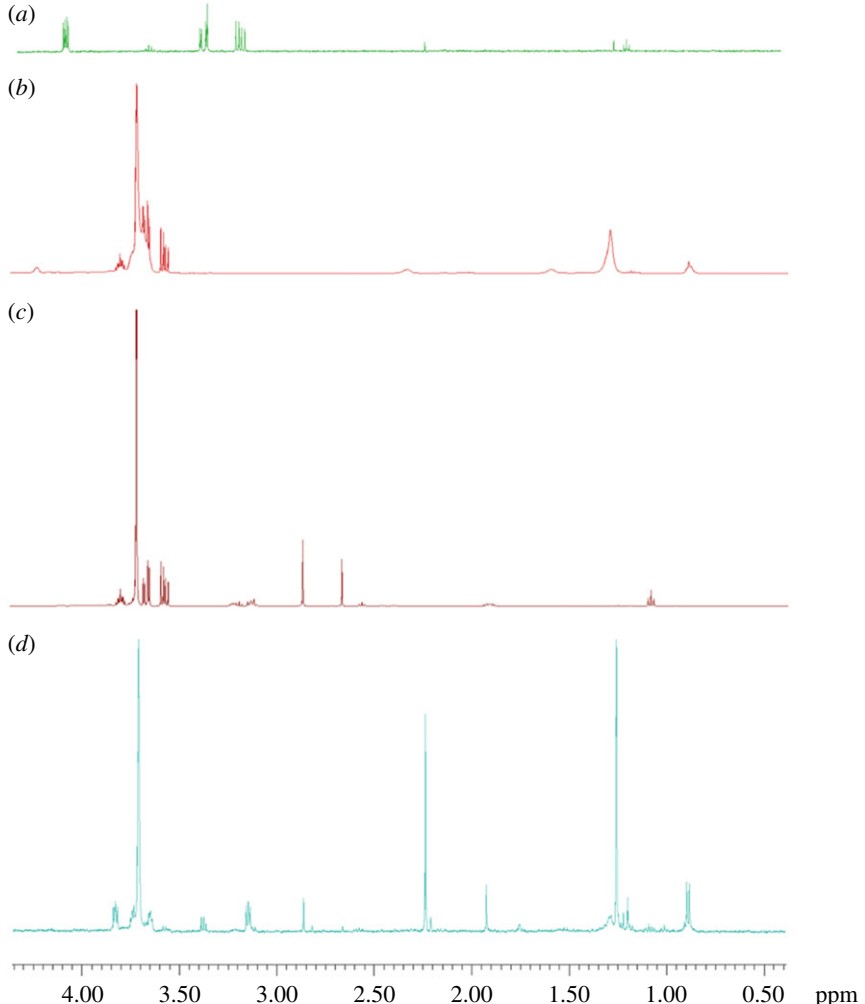

**Figure 1.** ¹H-NMR spectra of (a) L-cysteine, (b) cysteine-modified AuNSs, (c) NHS-terminated glucose oxidase and (d) AuNSs−Cys−GOx bioconjugates.

such as FTIR and C-NMR for structural elucidations employed in conjugation studies [1], and as done in some studies [40–42], were, however, not carried out in this study due to unavailability.

Figure 2 shows the UV–vis spectra of AuNSs functionalized with different ligands and their corresponding shifts in LSPR. The AuNSs that were modified with $10^{-6}$ M Cys had their LSPR at 716 nm compared to 712 nm for the control PVP-stabilized AuNSs. This denotes a slight red-shift, probably due to adsorption of Cys molecules on gold surface via the Au–S bond and a so-formed dielectric monolayer of thiol around AuNSs [18,43]. The LSPR for AuNSs−Cys−GOx bioconjugates was at 732 nm, showing a further red-shifting by 16 nm due to surface modification with $6.25 \times 10^{-4}$ g ml$^{-1}$ GOx. The red-shifting in the GOx-modified AuNSs is plausibly due to growth in size of the AuNSs after attaching to the enzyme. No broadening of the LPSR spectrum for GOx-modified AuNSs was observed which implied the maintenance of the structural integrity of the AuNSs after conjugation [18,44].

The morphology of the differently functionalized AuNSs was characterized by HR-TEM (figure 3). Figure 3a shows PVP-stabilized multi-branched gold nanostars. The cysteine-modified AuNSs (figure 3b) show AuNSs agglomeration possibly due to the formation of bonds between the surface-bound Cys molecules of adjacent AuNSs−Cys [39]. Figure 3c,d shows the AuNSs−Cys−GOx bioconjugates without and with staining with 1% silver nitrate at different magnifications. The TEM image of AuNSs shows good dispersity even after conjugation with GOx (electronic supplementary material, figure S1). The protein layer on the peripheral of the AuNSs surface could not be imaged without staining with 1% silver nitrate due to low electron resistance of protein molecules in HR-TEM examination [18]. However, the staining with silver nitrate enabled the visualization of some silver nanoparticles around the protein domain that were formed via reduction by the enzyme GOx [18]. Thus, the enzyme layer was observed as small dark spots of $3.12 \pm 0.08$ nm around the nanostars particles. Li *et al.* [18] reported similar

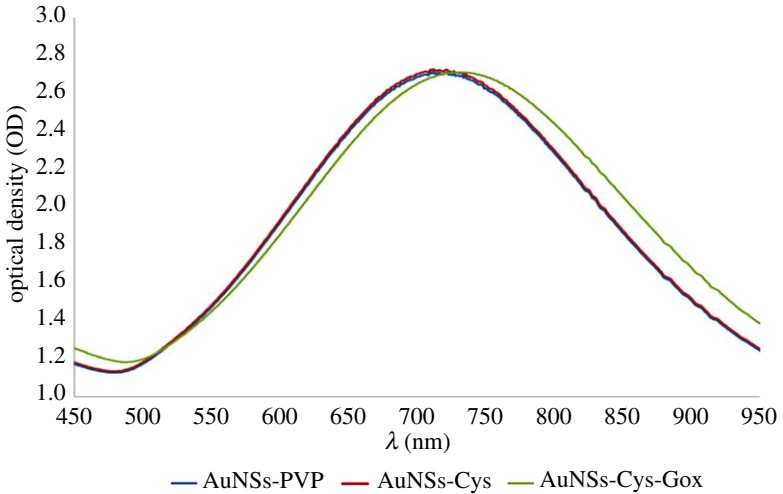

**Figure 2.** Normalized UV–vis spectra of PVP-stabilized AuNSs, cysteine-modified AuNSs and AuNSs–Cys–GOx bioconjugates.

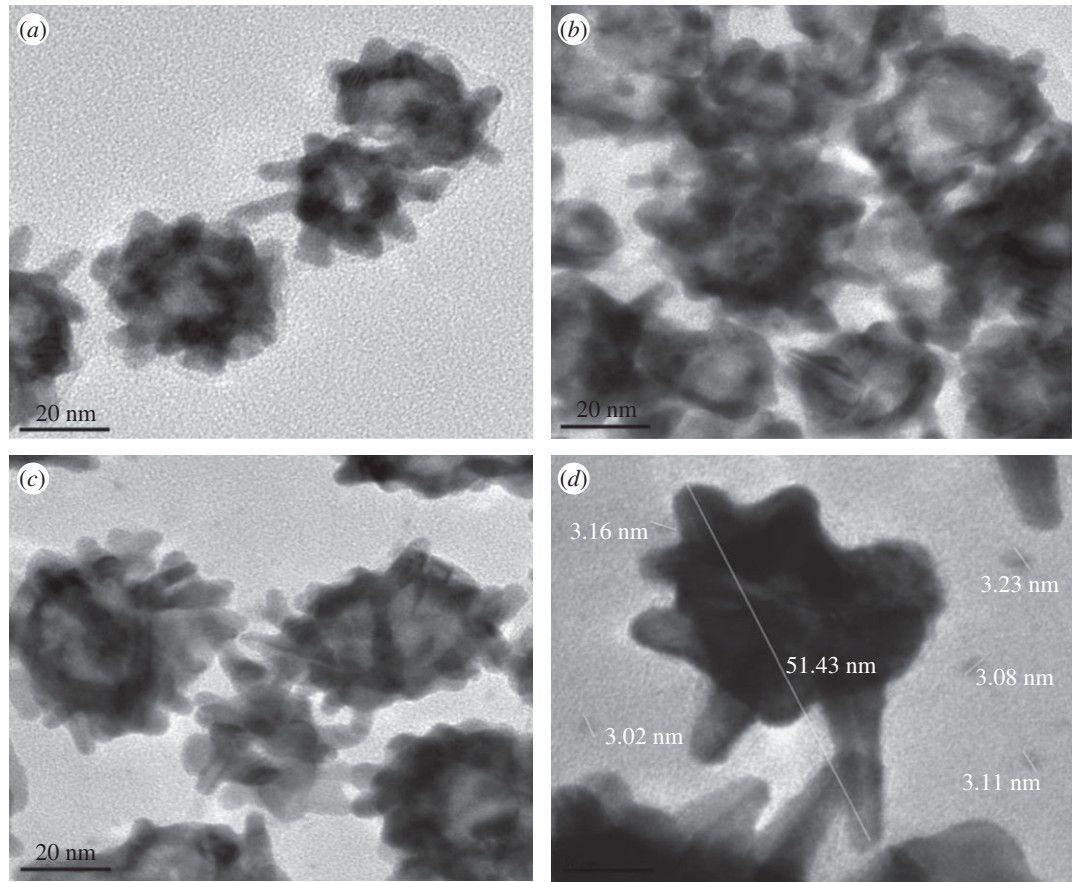

**Figure 3.** HR-TEM images of PVP-stabilized AuNSs (*a*), cysteine-modified AuNSs (*b*), AuNSs–Cys–GOx bioconjugates without staining (*c*) and AuNSs–Cys–GOx bioconjugates stained by 1% silver nitrate (*d*).

observations in visualizing the protein GOx after staining with silver nitrate. The AuNSs–Cys–GOx bioconjugates exhibited structural integrity and good dispersity in solution.

The stability of the fabricated AuNSs–Cys–GOx bioconjugates in solution was tested in solution of high ionic concentration. Figure 4 shows that there was no variation in the LSPR peaks by broadening and/or flattening of the AuNSs–Cys–GOx curve in salt compared to the one in ddH$_2$O. In comparison, the PVP-stabilized AuNSs had a slight decline in maximum absorbance by about 3% implying some loss of stability in the salt solution. This demonstrated the extra stability of the functionalized proteins added to the AuNSs relative to the PVP-stabilized AuNS. The application of

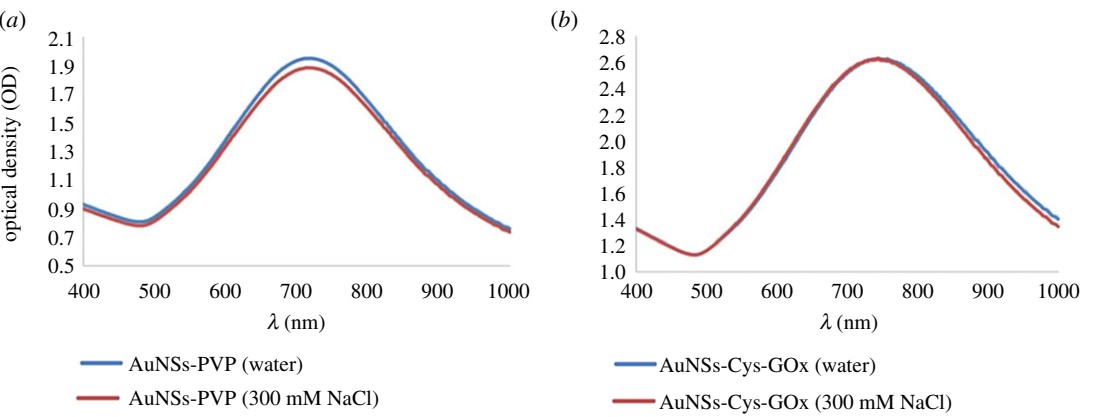

**Figure 4.** Comparison of UV–vis spectra of (*a*) PVP-stabilized AuNSs and (*b*) AuNSs–Cys–GOx bioconjugates in ddH$_2$O and 300 mM NaCl solutions for ionic stability tests.

**Figure 5.** Colorimetric photograph, UV–vis spectra of the mixture of 1 mM MES buffer (pH 6) and PVP-stabilized AuNSs in the presence of varying concentrations of glucose, and a plot of peak shifts versus glucose concentration.

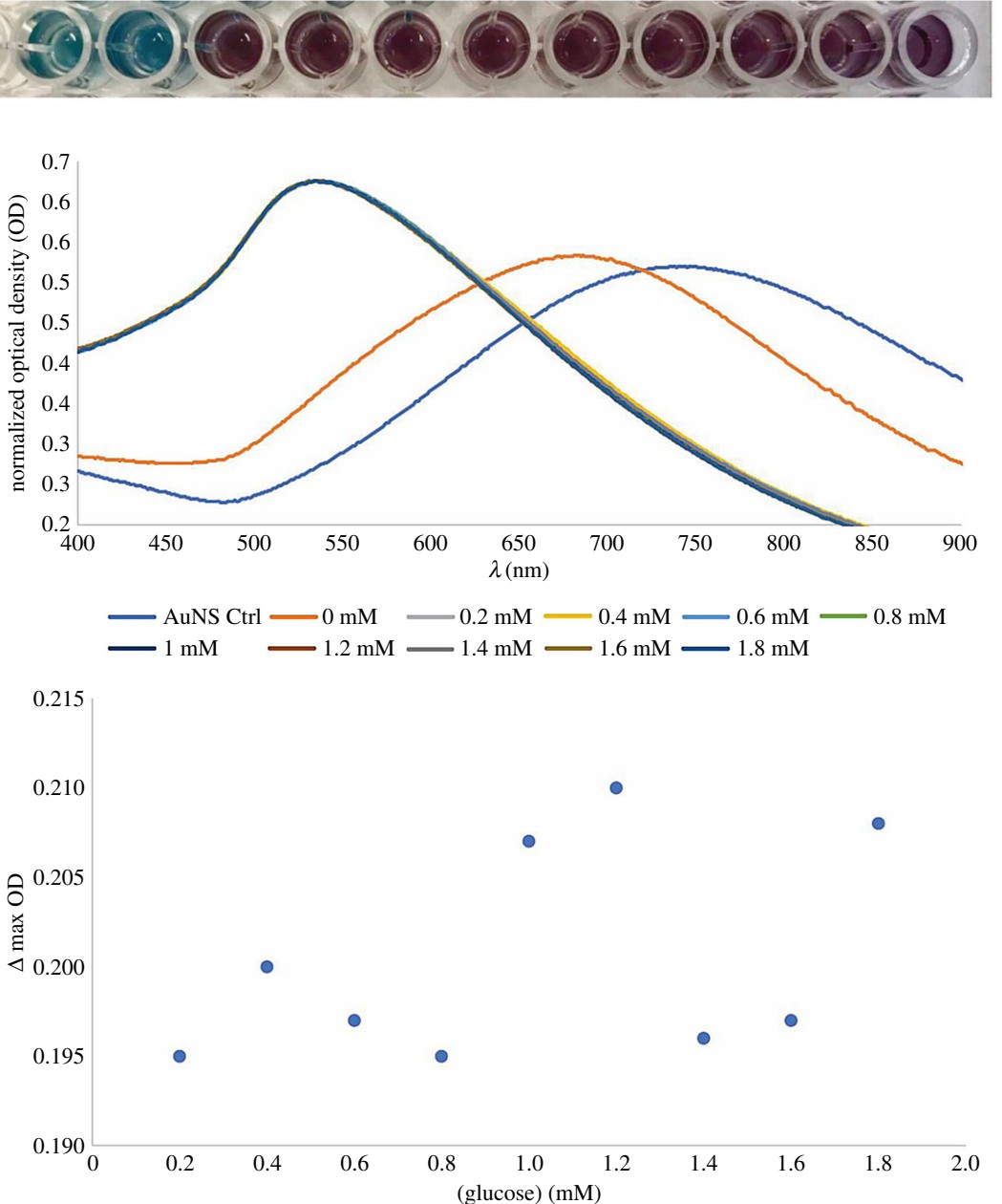

**Figure 6.** Colorimetric photograph, normalized UV–vis spectra of the mixture of 1 mM MES buffer (pH 6), 5 μl GOx solution and PVP-stabilized AuNSs in the presence of varying concentrations of glucose, and a plot of peak shifts versus glucose concentration.

functionalized nanostars in biological samples requires them to be stable in solutions containing high concentrations of proteins and salts [45]. If the AuNSs are not functionalized sufficiently, they are prone to aggregation in high ionic strength solutions in which the van der Waals attraction is stronger than the steric repulsion provided by the functionalization molecule [31,45].

The AuNSs–Cys–GOx bioconjugates were assessed as catalytic nanodevices in the oxidation of glucose and its plasmonic colorimetric sensing. The sensitivity and stability of the bioconjugates in glucose sensing was assessed based on the comparison assays. The mechanism for the sensing was enzyme-guided coating of silver ions onto the AuNSs surfaces. Figures 5–7 show the results of the comparison using differently functionalized AuNSs. Figure 5 shows the results for glucose sensing and detection using AuNSs without any GOx added to the reaction solutions. The AuNSs without any enzyme added to them could not distinguish between the different concentrations of glucose in solution. In fact, the AuNSs aggregated upon the adjustment of the pH to greater than 9 after the addition of detection solution as shown in the colorimetric photograph. The UV–vis spectra also confirmed this aggregation by the flattening of the absorption spectra in comparison to the control that had neither glucose nor detection solution added to it.

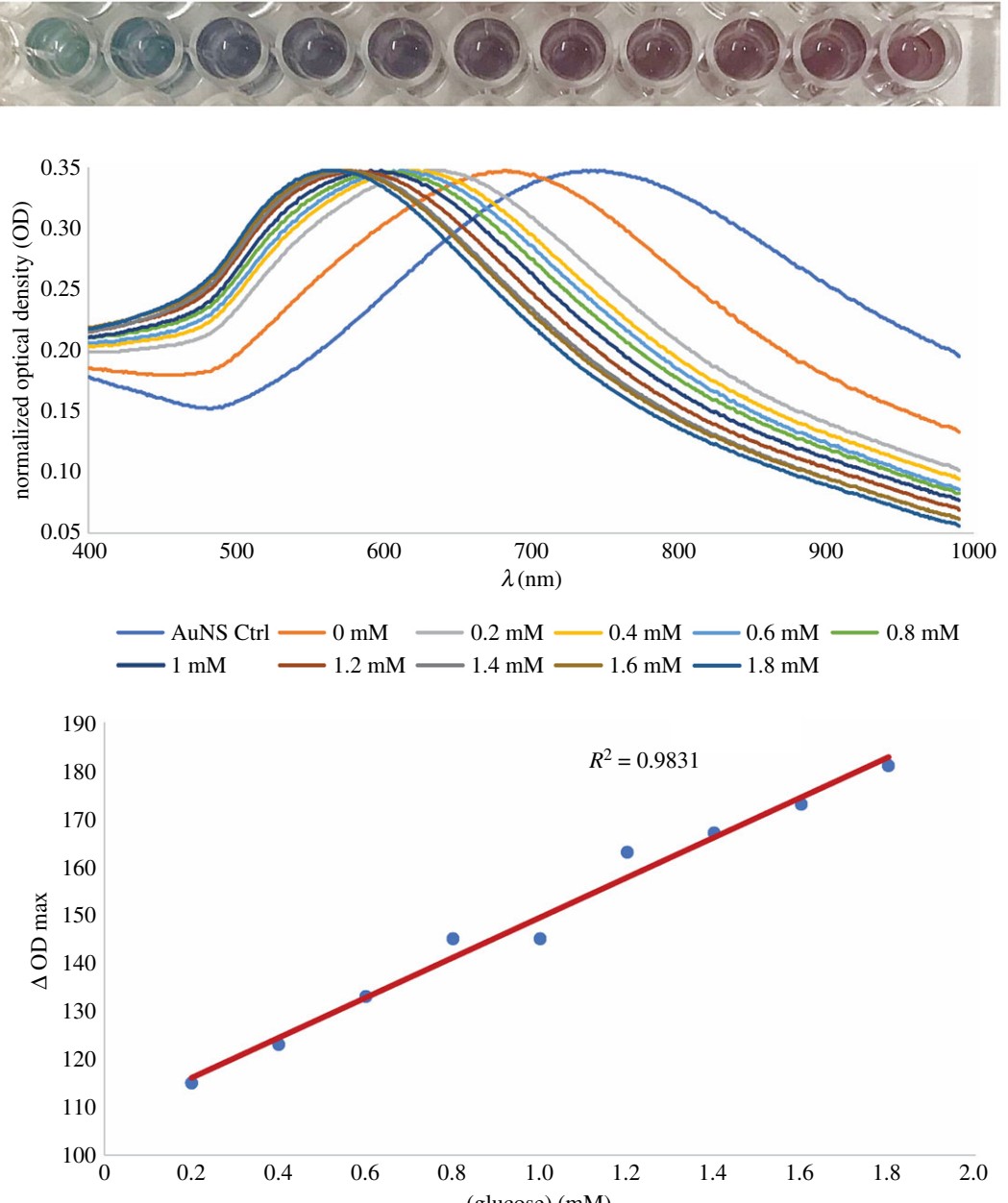

**Figure 7.** Colorimetric photograph of glucose assay, normalized UV – vis spectra of the mixture of 1 mM MES buffer (pH 6), 5 µl GOx solution and AuNSs – Cys – GOx bioconjugates in the presence of varying concentrations of glucose, and a plot of peak shifts versus glucose concentration.

Gold nanoparticles have been reported to act as nanozymes that mimic glucose oxidase in the oxidation of glucose [46,47]. It was clear from the observation made from this particular assay that AuNSs without sufficient functionalization with glucose oxidase were poor catalytic nanodevices for glucose sensing. Furthermore, poor stability was also observed in the presence of detection solution.

Figure 6 shows the assay with GOx added to the solutions in the reaction well. In this case, the GOx was attached to the AuNSs via physisorption. In comparison to the one assay without GOx added to it, there was observable colour change in response to the presence of glucose in the solution. The colour changed from blue to purple, but the assay could not discriminate distinctively, both colorimetrically and via LSPR peak shifts, between different concentrations of glucose in the solutions. As a result, there was no predictable correlation between signal generated and the varying concentrations of glucose. Physisorption is reportedly a poor method for enzyme immobilization as it tends to lead to covering of the active site onto the immobilization surface affecting the biorecognition and oxidation

of the target analyte [1,7]. This would explain why there were no discernible differences in the signal generated for different glucose concentrations.

Figure 7 shows the glucose assay using the AuNSs–Cys–GOx bioconjugates. The AuNSs–Cys–GOx bioconjugates showed distinguishable differences, both in colour and LSPR peak shifts, in response to different concentrations of glucose. The colours changed from blue to dark blue to purple and pink. The coefficient of determination between signal generated and concentration of glucose demonstrated a good model fit for the detection of glucose ($R^2 > 0.98$). The linear range was observed to be from 0.2 to 1.8 mM glucose. The limit of detection for this method was calculated to be 0.04 mM glucose. This showed that the AuNSs–Cys–GOx bioconjugates fabricated in the proposed approach had potential to be used for detection of low concentrations of analytes with greater sensitivity and stability compared to those of physisorption and bare AuNSs. With further optimizations in the conjugation process and the signal generation procedure, this could be used and extended to other enzymatic and antibody assays that utilize oxidases.

# 4. Conclusion

A bioconjugation approach for attachment of enzymes to gold nanostars is proposed that is simple and easy to replicate. This approach showed the ability to generate stable and sensitive AuNSs–Cys–GOx bioconjugates. The conjugation procedure potentially increased attachment affinity without protein adsorption onto the AuNSs surface by modifying the AuNSs with Cys. The production of an active ester intermediate on the enzyme using EDC/sulfo-NHS introduced a number of functional groups for covalent binding. The ratio of enzymes per nanoparticle was increased due to the many available functional groups between Cys and the NHS-terminated GOx for covalent attachment. As opposed to only functionalizing with EDC plus sulfo-NHS which is a zero-length cross-linker molecule, this approach enabled some relative separation distance between the enzyme and the AuNSs by the use of Cys, thus enabling attachment with the active site uncovered. The method produced AuNSs–Cys–GOx bioconjugates that maintained optimal function and activity for both the GOx and AuNSs. The produced AuNSs–Cys–GOx bioconjugates displayed greater sensitivity and stability in glucose sensing in comparison to the ones where the enzyme was simply added to the reaction well. All this demonstrated the potential the method has to fabricate AuNSs–enzyme bioconjugates for biosensing applications. Further studies are being pursued to replicate this bioconjugation strategy for immobilization of other oxidases in nanobiosensors.

Ethics. Ethical approval to carry out this study was granted by the North-West University Research Ethics Committee

Data accessibility. Data available from the Dryad Digital Repository at: https://doi.org/10.5061/dryad.95t9g4r [48].

Authors' contributions. M.M.P. conceived the study, designed the study, carried out the synthesis, modification and glucose testing laboratory work, participated in data analysis and drafted the manuscript; S.M. carried out the [1]H-NMR analysis, data analysis and drafting of the manuscript. B.C.V. and D.W.M. participated in design of the study, data analysis and drafting of the manuscript. All authors gave final approval for publication.

Competing interests. There are no competing interests to declare.

Funding. We are grateful for the funding from the North-West University's Centre of Human Metabolomics (CHM) and the South African Technology Innovation Agency (TIA) to carry out this work.

Acknowledgements. The authors would like to thank Dr Innocent Shuro from the Laboratory for Electron Microscopy, Chemical Resource Beneficiation, North-West University, Potchefstroom, South Africa, for assistance with nanoparticle characterization and imaging.

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
