## [Reviewer comments · Royal Society Open Science]

Review History

RSOS-190205.R0 (Original submission)

Review form: Reviewer 1

Is the manuscript scientifically sound in its present form?

Yes

Are the interpretations and conclusions justified by the results?

No

Is the language acceptable?

Yes

Is it clear how to access all supporting data?

Yes

Do you have any ethical concerns with this paper?

No

Have you any concerns about statistical analyses in this paper?

No

Recommendation?

Major revision is needed (please make suggestions in comments)

Comments to the Author(s)

The work by Masauso Moses Phiri et al has been subjected on the immobilisation of glucose oxidase onto gold nanostars to produce of AuNSs-Cys-GOx bioconjugate complex for increased attachment affinity, Good dispersity and Greater sensitivity of it. I think this method is not a facile method to immobilisation of glucose oxidase onto gold nanostars and this manuscript need to major and minor revision with following details:

Majors:

page 3 lines 3-4:

- "The PVP-stabilised seedless AuNSs were synthesized by reducing the H₂AuCl₄ with ascorbic acid and AgNO₃ added to control the growth of the branches based on the method reported by Phiri et al." The details of synthesis method should be clarified.

Page 3:

- NMR section is very detailed and should be summarised

Page 3 lines 53-54

- "Comparison assays were done using AuNSs without any GOx, AuNSs with unattached GOx in solution" what is difference between these two states?

- "The thiol moiety has been found to be a very effective site for interaction" how this with one S atom of Cys?

- The text of "Stability of AuNSs-Cys-GOx bioconjugates" section needs the reference.

- Where are the activity results?

Scheme 1 is very ambiguous:

- Where is roles of sulfo-NHS and EDC in scheme 1.

- What is the role of EDC in the AuNSs-Cys-GOx synthesis?

- Why in the GOx structure there are 2 NH₃⁺ groups?

- Why the arrow to belong from AuNS to sulfo-NHS?

- Where is the pvp in this scheme?

- Why the authors used the 2 linkers (NHS and EDC) in this work?

"The particle-size distribution was estimated by measuring the size of approximately 100 nanoparticles in different grid regions"

- How?

- Where is the figure including 100 NP and what was the software for this estimation?

- All of components in the TEM images should be recognized and assigned?

In Figure 1:

- the Roman numerals on the figure should be corrected.

- Zero point of the all spectra should be specified.

I do not understand:

- Spectrum (III) shows the peaks for the NHS-terminated GOx with sulfo-NHS in tandem with EDC.

In figure2:

- No difference between UV of AuNS-pvp and AuNS-Cys.

Page 5 line 60:

- I do not understand "No broadening of the GOx-modified AuNSs was observed which implies that the AuNSs maintained their structural integrity with good dispersity"

I do not understand:

- "How the Au nanostars formation was confirmed?"

- All the UV-Vis spectra 400-0 nm were scanned

- The figure 4 should be assigned and appeared consequently which explained in the text

- Why the intensity for AuNSs-Cys-GOx is higher than the AuNSs intensity?

- What is the goal of ionic strength investigation in this work?

- What are the Gox, enzyme and protein roles in this work?

- Generally the absorbance of AuNanoparticles have a max intensity before 600 nm wavelength but in this work this character has been appeared about 700 nm? Why?

- The figures in Fig 5 should be separate to some figures.

I cannot concluded this sentence:

- "This showed the potential of the AuNSs-Cys-GOx bioconjugates fabricated in the proposed approach to be used to detect low concentrations of analytes with high sensitivity and stability"

Minors:

- What is Km?

The in following mistakes in the text should be corrected:

"Colorimetric"

"by the use of intercalated"

"linkers of different lengths"

"is that is facile"

"energy-dispersive X-ray spectroscopy (EDS)"

Review form: Reviewer 2

Is the manuscript scientifically sound in its present form?

Yes

Are the interpretations and conclusions justified by the results?

No

Is the language acceptable?

Yes

Is it clear how to access all supporting data?

Not Applicable

Do you have any ethical concerns with this paper?

No

Have you any concerns about statistical analyses in this paper?

No

Recommendation?

Major revision is needed (please make suggestions in comments)

Comments to the Author(s)

This work synthesized gold nanostars based on the method reported in the literature. A stable AuNSs-Cys-GOx bioconjugate complex was obtained by modification with Cys and NHS-terminated glucose oxidase. The authors need to clearly establish how the work presented here is novel compared to the previous work. And some conclusive evidence is needed by addressing the below concerns:

1. Why using gold star? As I understand in your work, the L-cysteine (Cys) plays a role on binding glucose oxidase. Then why did authors use gold star? Just because of "The method is simple and easy to replicate"? Did authors compare the enzyme response on normal gold nanoparticles (not star-shaped) also?

Is the bare gold star bad for enzyme immobilization? If so, why?

If the stars nanostructures play an important role, then I will have a question: How about the influence of the number of branches of stars on enzyme immobilization?

2. Please explain how to avoid nonspecific binding of the protein with the authors' methods. And which data show the sample reducing nonspecific binding of the protein?

3. I suggest the authors to show SEM images to check the aggregation of the samples, because TEM images just showed a small range.

4. Except for NMR, it is necessary to run XPS characterization also to check the modification and immobilization.

Review form: Reviewer 3 (Raghu Anjanapura)

Is the manuscript scientifically sound in its present form?

Yes

Are the interpretations and conclusions justified by the results?

Yes

Is the language acceptable?

Yes

Is it clear how to access all supporting data?

Not Applicable

Do you have any ethical concerns with this paper?

No

Have you any concerns about statistical analyses in this paper?

I do not feel qualified to assess the statistics

Recommendation?

Major revision is needed (please make suggestions in comments)

Comments to the Author(s)

The article reports on Facile immobilisation of glucose oxidase onto gold nanostars with enhanced binding affinity and optimal function. It needs revision before accepting for the publications.

The purification of AuNSs-Cys-GOx bioconjugates is not clear. It should be taken care in the revised paper.

Hydrogen bonding plays a vital role in between NH & CO groups present in GOx to Cys-modified AuNSs nanoparticle. Hence, I author should include FTIR data and to confirm structure the following relevant below references can be consider.

Polymer Engineering & Science 54 (1), 24-32, 2014.

Journal of applied polymer science 104 (1), 81-88, 2007.

Journal of applied polymer science 106 (1), 299-308, 2007.

English and grammatical errors should be rectified during the revision of the paper.

Decision letter (RSOS-190205.R0)

12-Mar-2019

Dear Mr Phiri:

Title: Facile immobilisation of glucose oxidase onto gold nanostars with enhanced binding affinity and optimal function

Manuscript ID: RSOS-190205

The editor assigned to your manuscript has now received comments from reviewers. We would like you to revise your paper in accordance with the referee and Subject Editor suggestions which can be found below (not including confidential reports to the Editor). Please note this decision does not guarantee eventual acceptance.

Please submit your revised paper before 04-Apr-2019. Please note that the revision deadline will expire at 00.00am on this date. If we do not hear from you within this time then it will be assumed that the paper has been withdrawn. In exceptional circumstances, extensions may be possible if agreed with the Editorial Office in advance. We do not allow multiple rounds of revision so we urge you to make every effort to fully address all of the comments at this stage. If deemed necessary by the Editors, your manuscript will be sent back to one or more of the original reviewers for assessment. If the original reviewers are not available we may invite new reviewers.

To revise your manuscript, log into <http://mc.manuscriptcentral.com/rsos> and enter your Author Centre, where you will find your manuscript title listed under "Manuscripts with Decisions." Under "Actions," click on "Create a Revision." Your manuscript number has been

appended to denote a revision. Revise your manuscript and upload a new version through your Author Centre.

RSC Associate Editor:
Comments to the Author:
(There are no comments.)

RSC Subject Editor:
Comments to the Author:
(There are no comments.)

Reviewers' Comments to Author:
Reviewer: 1

Comments to the Author(s)

The work by Masauso Moses Phiri et al has been subjected on the immobilisation of glucose oxidase onto gold nanostars to produce of AuNSs-Cys-GOx bioconjugate complex for increased attachment affinity, Good dispersity and Greater sensitivity of it. I think this method is not a facile method to immobilisation of glucose oxidase onto gold nanostars and this manuscript need to major and minor revision with following details:

Majors:

page 3 lines 3-4:

"The PVP-stabilised seedless AuNSs were synthesized by reducing the H₂AuCl₄ with ascorbic acid and AgNO₃ added to control the growth of the branches based on the method reported by Phiri et al." The details of synthesis method should be clarified.

Page 3:

- NMR section is very detailed and should be summarised

Page 3 lines 53-54

- "Comparison assays were done using AuNSs without any GOx, AuNSs with unattached GOx in solution" what is difference between these two states?

- "The thiol moiety has been found to be a very effective site for interaction" how this with one S atom of Cys?

- The text of "Stability of AuNSs-Cys-GOx bioconjugates" section needs the reference.

- Where are the activity results?

Scheme 1 is very ambiguous:

- Where is roles of sulfo-NHS and EDC in scheme 1.
- What is the role of EDC in the AuNSs-Cys-GOx synthesis?
- Why in the GOx structure there are 2 NH₃⁺ groups?
- Why the arrow to belong from AuNS to sulfo-NHS?
- Where is the pvp in this scheme?
- Why the authors used the 2 linkers (NHS and EDC) in this work?

"The particle-size distribution was estimated by measuring the size of approximately 100 nanoparticles in different grid regions"

- How?
- Where is the figure including 100 NP and what was the software for this estimation?
- All of components in the TEM images should be recognized and assigned?

In Figure 1:

- the Roman numerals on the figure should be corrected.
- Zero point of the all spectra should be specified.

I do not understand:

- Spectrum (III) shows the peaks for the NHS-terminated GOx with sulfo-NHS in tandem with EDC.

In figure2:

- No difference between UV of AuNS-pvp and AuNS-Cys.

Page 5 line 60:

- I do not understand "No broadening of the GOx-modified AuNSs was observed which implies that the AuNSs maintained their structural integrity with good dispersity"

I do not understand:

- "How the Au nanostars formation was confirmed?"

- All the UV-Vis spectra 400-0 nm were scanned

- The figure 4 should be assigned and appeared consequently which explained in the text
- Why the intensity for AuNSs-Cys-GOx is higher than the AuNSs intensity?
- What is the goal of ionic strength investigation in this work?
- What are the Gox, enzyme and protein roles in this work?

- Generally the absorbance of AuNanoparticles have a max intensity before 600 nm wavelength but in this work this character has been appeared about 700 nm? Why?

- The figures in Fig 5 should be separate to some figures.

I cannot concluded this sentence:

- "This showed the potential of the AuNSs-Cys-GOx bioconjugates fabricated in the proposed approach to be used to detect low concentrations of analytes with high sensitivity and stability"

Minors:

- What is Km?

The in following mistakes in the text should be corrected:

"Colorimetric"

"by the use of intercalated"

"linkers of different lengths"

"is that is facile"

"energy-dispersive X-ray spectroscopy (EDS)"

Reviewer: 2

Comments to the Author(s)

This work synthesized gold nanostars based on the method reported in the literature. A stable AuNSs-Cys-GOx bioconjugate complex was obtained by modification with Cys and NHS-terminated glucose oxidase. The authors need to clearly establish how the work presented here is novel compared to the previous work. And some conclusive evidence is needed by addressing the below concerns:

1. Why using gold star? As I understand in your work, the L-cysteine (Cys) plays a role on binding glucose oxidase. Then why did authors use gold star? Just because of "The method is simple and easy to replicate"? Did authors compare the enzyme response on normal gold nanoparticles (not star-shaped) also?

Is the bare gold star bad for enzyme immobilization? If so, why?

If the stars nanostructures play an important role, then I will have a question: How about the influence of the number of branches of stars on enzyme immobilization?

2. Please explain how to avoid nonspecific binding of the protein with the authors' methods. And which data show the sample reducing nonspecific binding of the protein?

3. I suggest the authors to show SEM images to check the aggregation of the samples, because TEM images just showed a small range.

4. Except for NMR, it is necessary to run XPS characterization also to check the modification and immobilization.

Reviewer: 3

Comments to the Author(s)

The article reports on Facile immobilisation of glucose oxidase onto gold nanostars with enhanced binding affinity and optimal function. It needs revision before accepting for the publications.

The purification of AuNSs-Cys-GOx bioconjugates is not clear. It should be taken care in the revised paper.

Hydrogen bonding plays a vital role in between NH & CO groups present in GOx to Cys-modified AuNSs nanoparticle. Hence, I author should include FTIR data and to confirm structure the following relevant below references can be consider.

Polymer Engineering & Science 54 (1), 24-32, 2014.

Journal of applied polymer science 104 (1), 81-88, 2007.

Journal of applied polymer science 106 (1), 299-308, 2007.

English and grammatical errors should be rectified during the revision of the paper.

Author's Response to Decision Letter for (RSOS-190205.R0)

See Appendix A.

Decision letter (RSOS-190205.R1)

05-Apr-2019

Dear Mr Phiri:

Title: Facile immobilisation of glucose oxidase onto gold nanostars with enhanced binding affinity and optimal function

Manuscript ID: RSOS-190205.R1

It is a pleasure to accept your manuscript in its current form for publication in Royal Society Open Science. The chemistry content of Royal Society Open Science is published in collaboration with the Royal Society of Chemistry.

RSC Associate Editor
Comments to the Author:
(There are no comments.)

Reviewer(s)' Comments to Author:

Appendix A

Reviewer: 1

Comments to the Author(s)

The work by Masauso Moses Phiri et al has been subjected on the immobilisation of glucose oxidase onto gold nanostars to produce of AuNSs-Cys-GOx bioconjugate complex for increased attachment affinity, Good dispersity and Greater sensitivity of it. I think this method is not a facile method to immobilisation of glucose oxidase onto gold nanostars and this manuscript need to major and minor revision with following details:

Majors:

1. page 3 lines 3-4:

-"The PVP-stabilised seedless AuNSs were synthesized by reducing the H₂AuCl₄ with ascorbic acid and AgNO₃ added to control the growth of the branches based on the method reported by Phiri et al." The details of synthesis method should be clarified.

Response:

This has been done and now the section reads as follows in the text:

A recently published method for the synthesis of seedless silver and ascorbic acid assisted AuNSs by Phiri *et al.* (Phiri *et al.*, 2019) was followed. Briefly, 10 mL of ddH₂O was acidified with 10 µL of 1M HCl. Thereafter, 50 µL of 100 mM ascorbic acid was added under mild stirring. 50 µL of 50 mM H₂AuCl₄ was added to the mixture. Shortly and rapidly (within 30 seconds), 50 µL of 10mM AgNO₃ was added to the solution which resulted in a deep blue colour change within a few seconds. Finally, 500 µL of 2.5 mM PVP to the mixture. The prepared AuNSs were cleaned-up by centrifugation for 90 minutes at 3000g. The pellet was then recovered and re-suspended in 2 mL of ddH₂O.

2. Page 3:

- NMR section is very detailed and should be summarized

Response:

The above-mentioned section has since been summarised and reads as follows:

¹H-NMR analyses of the samples in fabrication stages was done according to the method by Venter *et al* (Venter *et al.*, 2018). 600 µL of samples were measured at 500MHz on a Bruker Avance III HD NMR spectrometer equipped with a triple-resonance inverse (TXI) 1H[15N,13C] probe head and x, y, z gradient coils. ¹H spectra were acquired as 128 transients in 32K data points with a spectral width of 12002 Hz. Fourier transformation and phase and base line correction were done automatically. Software used for NMR processing was Bruker Topspin (V3.5). Bruker AMIX (V3.9.14) was used for metabolite identification (Ellinger *et al.*, 2013).

3. Page 3 lines 53-54

- "Comparison assays were done using AuNSs without any GOx, AuNSs with unattached GOx in solution" what is difference between these two states?

Response:

This section has been rewritten in order to clarify the difference between the two states. The following is an extract from the section:

Three comparison assays were done to assess if the AuNSs-Cys-GOx bioconjugation method offered any advantage in biosensing; 1) AuNSs only without the addition of any GOx to it as assay controls. 2) AuNSs with 5 µL GOx added to the reaction solutions. 3) Lastly, AuNSs-Cys-GOx bioconjugates in solution for both oxidation and detection of glucose.

4. - "The thiol moiety has been found to be a very effective site for interaction" how this with one S atom of Cys?

Response:

Reference is made to the two following publications, listed below, that explain this in greater detail. In brief: Zhao et al., performed a comprehensive analysis of cysteine–gold cluster complexes. In contrast to the conventional understanding of the covalent nature of the S–Au bond, using computational methods, they present that the S–Au bond exhibits both covalent and donor–acceptor characters. One stable isomer of Au³–Cys^S was specially designed to demonstrate these two bonding components explicitly. They concluded that generally, the bonding strength between gold clusters and cysteine is positively correlated with the S–Au overlap-weighted bond order, but negatively correlated with the S–Au bond length.

1. Zhao, Y., Zhou, F., Zhou, H., Su, H. 2013 The structural and bonding evolution in cysteine–gold cluster complexes. *Physical Chemistry Chemical Physics*. **15**, 1690-1698.
 2. Majzik, A., Fülöp, L., Csapó, E., Bogár, F., Martinek, T., Penke, B., Bíró, G., Dékány, I. 2010 Functionalization of gold nanoparticles with amino acid, β -amyloid peptides and fragment. *Colloids and Surfaces B: Biointerfaces*. **81**, 235-241.
5. - The text of "Stability of AuNSs-Cys-GOx bioconjugates" section needs the reference.

Response:

The section has been rewritten and citation included in the manuscript.

6. - Where are the activity results?

Response:

The heading on page 3 line 49 has been modified to "Glucose sensing using AuNSs-Cys-GOx bioconjugates" so that it is not misleading to the reader what exactly was done.

7. Scheme 1 is very ambiguous:

- i. Where is roles of sulfo-NHS and EDC in scheme 1.
- ii. What is the role of EDC in the AuNSs-Cys-GOx synthesis?
- iii. Why in the GOx structure there are 2 NH₃⁺ groups?
- iv. Why the arrow to belong from AuNS to sulfo-NHS?
- v. Where is the pvp in this scheme?

Response:

Scheme 1 shows a summarized version of the whole reaction after the removal of excess PVP. The remaining PVP is functionalised to the surface of the AuNSs and some is removed via ligand exchange with L-cysteine.

All the structures are all schematic illustrations of the plausible reactions. For example, the 2 NH₃⁺ groups on the enzymes are for illustrative purposes only, and not an exact representation of the enzyme.

The roles of EDC in the scheme is to react with a carboxylate group to form an active ester leaving group. The advantage of adding sulfo-NHS to EDC reactions is to increase the solubility and stability of the active intermediate, which ultimately reacts with the attacking amine. EDC/sulfo-NHS coupled reactions are highly efficient and tend to increase the yield of conjugation significantly over that obtainable solely with EDC.

The arrow from the L-cysteine-modified AuNSs crisscrossed with the one from the sulfo-NHS activated intermediate illustrating the reaction of the active ester being mixed with amine-containing AuNSs for conjugation. The two-step process of first incubating the enzyme with EDC/sulfo-NHS allows the formation of the active species only on the enzyme, thus gaining control over the conjugation reaction.

8. Why the authors used the 2 linkers (NHS and EDC) in this work?

Response:

EDC reacts with a carboxylate group to form an active ester leaving group. The advantage of adding sulfo-NHS to EDC reactions is to increase the solubility

and stability of the active intermediate, which ultimately reacts with the attacking amine. EDC/sulfo-NHS coupled reactions are highly efficient and tend to increase the yield of conjugation significantly over that obtainable solely with EDC (Hermanson, 2013).

9. "The particle-size distribution was estimated by measuring the size of approximately 100 nanoparticles in different grid regions"

- How?

- Where is the figure including 100 NP and what was the software for this estimation?

- All of components in the TEM images should be recognized and assigned?

Response:

In the revised manuscript, the authors have removed this part from the methodology so as to narrow down to the main focus of the study – the immobilisation of enzymes onto the nanoparticles.

10. In Figure 1:

- the Roman numerals on the figure should be corrected.

Response:

This has since been corrected.

- Zero point of the all spectra should be specified.

Response:

Only spectral regions of differentiation between L-Cysteine, Cysteine-modified AuNSs, NHS-terminated glucose oxidase, and AuNSs-Cys-GOx bioconjugates is shown. This is in accordance with the accepted norm of reported NMR data.

11. I do not understand:

- Spectrum (III) shows the peaks for the NHS-terminated GOx with sulfo-NHS in tandem with EDC.

Response:

The sentence has been modified to read as: "Spectrum (III) shows the peaks for the NHS-terminated GOx."

12. In figure2:

- No difference between UV of AuNS-pvp and AuNS-Cys.

Response:

There was a slight red-shift from 712 nm to 716 nm after the addition of 10^{-6} M L-cysteine. Using both the NMR and UV-vis spectroscopy characterisation, a good confirmation can be stated this shift is likely from the absorption of L-cysteine molecules on gold nanostars surface.

13. Page 5 line 60:

- I do not understand "No broadening of the GOx-modified AuNSs was observed which implies that the AuNSs maintained their structural integrity with good dispersity"

Response:

The sentence has been corrected and now reads as: "No broadening of the LPSR spectra for GOx-modified AuNSs was observed which implies that the AuNSs maintained their structural integrity with good dispersity."

14. I do not understand:

- "How the Au nanostars formation was confirmed?"

Response:

The UV-vis spectra of the synthesized AuNSs were observed to have typical LSPRs of star-shaped nanoparticles as judged by the longer wavelengths and

the broad peaks. Increased aspect ratios of the branches make the longitudinal components of the plasmon band become more intense and red-shifted relative to the LSPR of spherical particles (Maiorano *et al.*, 2011; Rodríguez-Lorenzo *et al.*, 2012). The TEM images showed the morphologies of the nanocrystals to have star-shaped or multi-branched (as alternatively referred to) (Maiorano *et al.*, 2011; Li *et al.*, 2013; Chirico *et al.*, 2015; de Puig *et al.*, 2015).

- All the UV-Vis spectra 400-900 nm were scanned

Response:

This was corrected and now reads as follows: "All the UV-vis spectra scans were from 400 – 990 nm."

- The figure 4 should be assigned and appeared consequently which explained in the text

Response:

The exact request here is however, not fully understood by us. The position of the figure relative to the text has nonetheless been adjusted.

- Why the intensity for AuNSs-Cys-GOx is higher than the AuNSs intensity?

Response:

The UV-vis spectra in Figure 4 are not normalised. With the functionalisation of the AuNSs with GOx there is an increase in the absorbance.

- What is the goal of ionic strength investigation in this work?

Response:

Literature requires that the functionalized nanoparticles are tested for stability using a flocculation assay to investigate their ability to be used in biological fluids (Wangoo *et al.*, 2008). Application of functionalized nanostars in biological samples requires them to be stable in solutions containing high concentrations of proteins and salts (Wangoo *et al.*, 2008). If the AuNSs are

not functionalized sufficiently, they are prone to aggregation in high ionic strength solutions in which the van der Waals attraction is stronger than the steric repulsion provided by the functionalization molecule (Wangoo *et al.*, 2008; Rodríguez-Lorenzo *et al.*, 2012).

- What are the GOx, enzyme and protein roles in this work?

Response:

The GOx – which is both a protein and enzyme in this case, provides surface functionalisation that offers stability to the gold nanoparticles – in our case, gold nanostars (Wangoo *et al.*, 2008).

- Generally, the absorbance of Au Nanoparticles have a max intensity before 600 nm wavelength but in this work this character has been appeared about 700 nm? Why?

Response:

AuNSs colloids display a wide and distinct LSPR feature, including an intense band typically centred around 650–900 nm. Below are a few references in that regard:

- Chirico, G., Borzenkov, M. & Pallavicini, P. 2015. Gold Nanostars: Synthesis, Properties and Biomedical Application: Springer.
- Guerrero-Martínez, A., Barbosa, S., Pastoriza-Santos, I. & Liz-Marzán, L.M. 2011. Nanostars shine bright for you: colloidal synthesis, properties and applications of branched metallic nanoparticles. *Current Opinion in Colloid & Interface Science*, 16(2):118-127.
- Guo, Y., Wu, J., Li, J. & Ju, H. 2016. A plasmonic colorimetric strategy for biosensing through enzyme guided growth of silver nanoparticles on gold nanostars. *Biosens Bioelectron*, 78:267-273.
- Maiorano, G., Rizzello, L., Malvindi, M.A., Shankar, S.S., Martiradonna, L., Falqui, A., Cingolani, R. & Pompa, P.P. 2011. Monodispersed and size-controlled multibranch gold nanoparticles with nanoscale tuning of surface morphology. *Nanoscale*, 3(5):2227-2232.

- Rodríguez-Lorenzo, L., De La Rica, R., Álvarez-Puebla, R.A., Liz-Marzán, L.M. & Stevens, M.M. 2012b. Plasmonic nanosensors with inverse sensitivity by means of enzyme-guided crystal growth. *Nature materials*, 11(7):604-607.
- Saverot, S., Geng, X., Leng, W., Vikesland, P., Grove, T. & Bickford, L. 2016. Facile, tunable, and SERS-enhanced HEPES gold nanostars. *RSC Advances*, 6(35):29669-29673.

- The figures in Fig 5 should be separate to some figure.

Response:

Again, we did not fully understand the comment here. However, the figures have since been presented as separate figures.

I cannot conclude this sentence:

- "This showed the potential of the AuNSs-Cys-GOx bioconjugates fabricated in the proposed approach to be used to detect low concentrations of analytes with high sensitivity and stability"

Response:

The section has been rewritten in order to increase the clarity.

Minors:

- What is K_m ? The Michaelis constant (K_m)

The in following mistakes in the text should be corrected:

"Colorimetric" Corrected

"by the use of intercalated" Corrected

"linkers of different lengths" Corrected

"is that is facile" Corrected

"energy-dispersive X-ray spectroscopy (EDS)" Corrected

All the minor corrections have been dealt with accordingly in the manuscript.

Reviewer: 2

Comments to the Author(s)

This work synthesized gold nanostars based on the method reported in the literature. A stable AuNSs-Cys-GOx bioconjugate complex was obtained by modification with Cys and NHS-terminated glucose oxidase. The authors need to clearly establish how the work presented here is novel compared to the previous work. And some conclusive evidence is needed by addressing the below concerns:

1. Why using gold star? As I understand in your work, the L-cysteine (Cys) plays a role on binding glucose oxidase. Then why did authors use gold star? Just because of "The method is simple and easy to replicate"? Did authors compare the enzyme response on normal gold nanoparticles (not star-shaped) also?

Is the bare gold star bad for enzyme immobilization? If so, why?

If the stars nanostructures play an important role, then I will have a question: How about the influence of the number of branches of stars on enzyme immobilization?

Response:

Gold nanostars were used mostly for two reasons; (i) signal transduction, and (ii) larger surface area for enzyme immobilisation.

Signal transduction is based on the localised surface plasmon resonance (LSPR) of the nanoparticles. The width, position, and number of LSPRs are

also determined by the shape of the Au NPs (Xia & Halas, 2005). A common feature of LSPRs for nanostars is that they are located at lower energy compared to the spheres. For example, Au nanospheres with the size of 2–50 nm show only one plasmon band centred at about 520 nm, while for nanostars the plasmon band is redshifted and more intense (Amendola *et al.*, 2017). The combination of dielectric properties and chemical stability of nanostars make them ideal substrates for the study of well-defined localized surface plasmon resonances (LSPRs) within the visible and near-infrared spectral range (Guerrero-Martínez *et al.*, 2011). Gold nanostars are easier to tune in a biochemical process for LSPR shifting either by their growth in size or change in shape to more a spherical morphology.

Secondly, nanostars being made from Au, act as a very efficient support matrix for enzyme immobilisation (Ahmad & Sardar, 2015). Nanostars are usually larger in size than nanospheres thereby providing a larger surface areas for enzyme immobilisation with potential for higher load of enzymes per nanoparticle (Sapsford *et al.*, 2013).

The bare star is not bad for functionalisation. The aim for the addition of cysteine was help with attaining some of the ideal criteria for ideal immobilisation such as having control over relative separation distance from the nanostar; control over enzyme orientation by preventing the enzyme's active site from being covered if it absorbs on the surface of the nanostar; enhance the attachment affinity of the enzyme to the nanostars. In short, the aim was an attempt at finding optimal enzyme immobilisations strategies to our synthesised gold nanostars.

The attachment approach that we used in this study was a covalent attachment to the surface of L-cysteine-modified nanostars. In this case, we postulate that the branches have no effect on the immobilisation process.

2. Please explain how to avoid nonspecific binding of the protein with the authors' methods. And which data show the sample reducing nonspecific binding of the protein?

Response:

Experimentally, it has been reported that functionalisation of gold nanoparticles with thiolated ligands leads to the formation of self-assembled monolayers of thiols on the gold surfaces that has the potential to reduce nonspecific protein absorption drastically (Lahiri *et al.*, 1999). Thus, this study was building on such literature studies that have leveraged this knowledge to functionalise proteins in such a way as to reduce the potential for nonspecific absorption (Li *et al.*, 2007; Pandey *et al.*, 2007; Bezbradica *et al.*, 2014). For example, cysteine was used together with glutaraldehyde to functionalise enzymes on an epoxy-activated support. Mercaptoundecanoic acid (MUA) has is reported to have been used to optimise enzyme immobilisation with thermal stability. It is therefore expected that there was no unspecific protein absorption based on the addition of cysteine that typically has the tendency to prevent it.

3. I suggest the authors to show SEM images to check the aggregation of the samples, because TEM images just showed a small range.

Response:

SEM images were not taken at the time of imaging but TEM images with lower magnifications covering a number of AuNSs were taken during microscopic characterisation of the glucose oxidase – modified AuNSs. Such an image has since been added as supplementary information.

4. Except for NMR, it is necessary to run XPS characterization also to check the modification and immobilization.

Response:

Noted with thanks. Although this technique is currently unavailable to us, it could in future be done in collaboration with others.

Reviewer: 3

Comments to the Author(s)

The article reports on Facile immobilisation of glucose oxidase onto gold nanostars with enhanced binding affinity and optimal function. It needs revision before accepting for the publications.

The purification of AuNSs-Cys-GOx bioconjugates is not clear. It should be taken care in the revised paper.

Response:

The section has been adjusted accordingly to include those details.

Hydrogen bonding plays a vital role in between NH & CO groups present in GOx to Cys-modified AuNSs nanoparticle. Hence, I author should include FTIR data and to confirm structure the following relevant below references can be consider.

Polymer Engineering & Science 54 (1), 24-32, 2014.

Journal of applied polymer science 104 (1), 81-88, 2007.

Journal of applied polymer science 106 (1), 299-308, 2007.

Response:

FTIR technique was our method-of-choice but was unfortunately not available to us at the time of characterisation. Hence, we resorted to using various techniques such as the NMR, UV-vis spectroscopy, microscopy, and dynamic light scattering (although the data was not included in the manuscript). However, we shall try to include FTIR in our on-going work of functionalisation of enzymes to nanoparticles.

The suggested references are well appreciated and appropriately referred to in the manuscript.

English and grammatical errors should be rectified during the revision of the paper.

This has also been dealt with accordingly.

Cited Works

1. Ahmad, R. & Sardar, M. 2015. Enzyme immobilization: an overview on nanoparticles as immobilization matrix. *Biochemistry and Analytical Biochemistry*, 4(2):1.
2. Aldewachi, H., Chalati, T., Woodroffe, M., Bricklebank, N., Sharrack, B. & Gardiner, P. 2018. Gold nanoparticle-based colorimetric biosensors. *Nanoscale*, 10(1):18-33.
3. Amendola, V., Pilot, R., Frascioni, M., Marago, O.M. & Iati, M.A. 2017. Surface plasmon resonance in gold nanoparticles: a review. *Journal of Physics: Condensed Matter*, 29(20):203002.
4. Bezbradica, D.I., Mateo, C. & Guisan, J.M. 2014. Novel support for enzyme immobilization prepared by chemical activation with cysteine and glutaraldehyde. *Journal of Molecular Catalysis B: Enzymatic*, 102:218-224.
5. Chirico, G., Borzenkov, M. & Pallavicini, P. 2015. Gold Nanostars: Synthesis, Properties and Biomedical Application: Springer.
6. de Puig, H., Tam, J.O., Yen, C.-W., Gehrke, L. & Hamad-Schifferli, K. 2015. Extinction coefficient of gold nanostars. *The Journal of Physical Chemistry C*, 119(30):17408-17415.
7. Ellinger, J.J., Chylla, R.A., Ulrich, E.L. & Markley, J.L. 2013. Databases and software for NMR-based metabolomics. *Current Metabolomics*, 1(1):28-40.
8. Guerrero-Martínez, A., Barbosa, S., Pastoriza-Santos, I. & Liz-Marzán, L.M. 2011. Nanostars shine bright for you: colloidal synthesis, properties and applications of branched metallic nanoparticles. *Current Opinion in Colloid & Interface Science*, 16(2):118-127.
9. Hermanson, G.T. 2013. Zero-length crosslinkers. Bioconjugate techniques. 3 ed. London, UK: Academic Press. p. 259-266).
10. Lahiri, J., Isaacs, L., Tien, J. & Whitesides, G.M. 1999. A strategy for the generation of surfaces presenting ligands for studies of binding based on

an active ester as a common reactive intermediate: a surface plasmon resonance study. *Analytical Chemistry*, 71(4):777-790.

11. Li, D., He, Q., Cui, Y., Duan, L. & Li, J. 2007. Immobilization of glucose oxidase onto gold nanoparticles with enhanced thermostability. *Biochemical and biophysical research communications*, 355(2):488-493.
12. Li, Y., Ma, J. & Ma, Z. 2013. Synthesis of gold nanostars with tunable morphology and their electrochemical application for hydrogen peroxide sensing. *Electrochimica Acta*, 108:435-440.
13. Maiorano, G., Rizzello, L., Malvindi, M.A., Shankar, S.S., Martiradonna, L., Falqui, A., Cingolani, R. & Pompa, P.P. 2011. Monodispersed and size-controlled multibranching gold nanoparticles with nanoscale tuning of surface morphology. *Nanoscale*, 3(5):2227-2232.
14. Pandey, P., Singh, S.P., Arya, S.K., Gupta, V., Datta, M., Singh, S. & Malhotra, B.D. 2007. Application of thiolated gold nanoparticles for the enhancement of glucose oxidase activity. *Langmuir*, 23(6):3333-3337.
15. Phiri, M.M., Mulder, D.W. & Vorster, B.C. 2019. Seedless gold nanostars with seed-like advantages for biosensing applications. *Royal Society Open Science*, 6(2):181971.
16. Rodríguez-Lorenzo, L., De La Rica, R., Álvarez-Puebla, R.A., Liz-Marzán, L.M. & Stevens, M.M. 2012. Plasmonic nanosensors with inverse sensitivity by means of enzyme-guided crystal growth. *Nature materials*, 11(7):604.
17. Sapsford, K.E., Algar, W.R., Berti, L., Gemmill, K.B., Casey, B.J., Oh, E., Stewart, M.H. & Medintz, I.L. 2013. Functionalizing nanoparticles with biological molecules: developing chemistries that facilitate nanotechnology. *Chemical reviews*, 113(3):1904-2074.
18. Venter, L., Mienie, L.J., van Rensburg, P.J.J., Mason, S., Vosloo, A. & Lindeque, J.Z. 2018. The cross-tissue metabolic response of abalone (*Haliotis midae*) to functional hypoxia. *Biology open*, 7(3):bio031070.
19. Wangoo, N., Bhasin, K.K., Mehta, S.K. & Suri, C.R. 2008. Synthesis and capping of water-dispersed gold nanoparticles by an amino acid: bioconjugation and binding studies. *J Colloid Interface Sci*, 323(2):247-254.

20. Xia, Y. & Halas, N.J. 2005. Shape-controlled synthesis and surface plasmonic properties of metallic nanostructures. *MRS bulletin*, 30(5):338-348.